# Circulating hypervirulent Marek's disease viruses in vaccinated chicken flocks in Taiwan by genetic analysis of *meq* oncogene

**Ming-Chu Cheng**[1,2,3]* , **Guan-Hua Lai**[1,2,3] , **Yi-Lun Tsai**[1] , **Yi-Yang Lien**[1,2,3]

**1** Department of Veterinary Medicine, College of Veterinary Medicine, National Pingtung University of Science and Technology, Pingtung, Taiwan, **2** Animal Disease Diagnostic Center, National Pingtung University of Science and Technology, Pingtung, Taiwan, **3** Research Center of Animal Biologics, National Pingtung University of Science and Technology, Pingtung, Taiwan

☯ These authors contributed equally to this work.
\* mccheng@mail.npust.edu.tw

**Data Availability Statement:** The novel meq oncogene sequences generated in this study were submitted to the GenBank database (accession numbers OQ576796-OQ576813).

## Abstract

Marek's disease (MD) is an important neoplastic disease caused by serotype 1 Marek's disease virus (MDV-1), which results in severe economic losses worldwide. Despite vaccination practices that have controlled the MD epidemic, current increasing MD-suspected cases indicate the persistent viral infections circulating among vaccinated chicken farms in many countries. However, the lack of available information about phylogeny and molecular characterization of circulating MDV-1 field strains in Taiwan reveals a potential risk in MD outbreaks. This study investigated the genetic characteristics of 18 MDV-1 strains obtained from 17 vaccinated chicken flocks in Taiwan between 2018 and 2020. Based on the sequences of the *meq* oncogene, the phylogenetic analysis demonstrated that the circulating Taiwanese MDV-1 field strains were predominantly in a single cluster that showed high similarity with strains from countries of the East Asian region. Because the strains were obtained from CVI988/Rispens vaccinated chicken flocks and the molecular characteristics of the Meq oncoprotein showed features like vvMDV and vv+MDV strains, the circulating Taiwanese MDV-1 field strains may have higher virulence compared with vvMDV pathotype. In conclusion, the data presented demonstrates the circulation of hypervirulent MDV-1 strains in Taiwan and highlights the importance of routine surveillance and precaution strategies in response to the emergence of enhanced virulent MDV-1.

## Introduction

Marek's disease (MD), caused by *Gallid alphaherpesvirus* 2 (GaHV-2), is a critical, highly contagious avian viral disease that induces serial clinical manifestations including systemic visceral lymphoma, neurological disorders, paralysis, and immunosuppression in infected chickens, resulting in considerable economic losses in poultry industry [1,2]. The etiological agent GaHV-2, also commonly known as serotype 1 of Marek's disease virus (MDV-1), belongs to a member of the genus *Mardivirus* in the subfamily *Alphaherpesvirinae* of the family *Herpesviridae*, which also consists of other non-oncogenic MDV species: *Gallid alphaherpesvirus 3*

**Funding:** The author(s) received no specific funding for this work.

**Competing interests:** The authors have declared that no competing interests exist.

(GaHV-3) is serotype 2 of MDV (MDV-2), and *Meleagrid alphaherpesvirus* 1, also known as turkey herpesvirus (HVT), is serotype 3 [3]. Nononcogenic MDVs were developed as first-generation vaccines and were soon after introduced to many countries for MD prevention [4]. Based on the pathotyping protocol referring to the virulent properties in surmounting specific vaccinal protection, the population of MDV-1 can be classified into various pathotypes, including mild (m), virulent (v), very virulent (vv), and very virulent plus (vv+) [5]. The increasing emergence of MD cases has been revealed in current reports among vaccinated chicken flocks in many countries, which suggests a probable rise in evolved MDV-1 field strains associated with enhanced virulence [6,7].

The MDV-1 genome encodes more than 200 genes, some of which are unique oncogenes primarily involved in viral pathogenesis [8]. The Meq oncoprotein encoded by the *meq* oncogene was the first discovered oncoprotein whose N-terminal basic-leucine-zipper (bZIP) domain and C-terminal proline-rich transactivation domain were identified as major functional factors associated with MDV-1 virulence and oncogenicity [9]. Recent studies have reported that specific amino acid mutations, proline contents, and the number of 4-proline-repeat stretches (PPPPs) within Meq oncoprotein, are correlated with MDV-1 virulence [10,11]. Therefore, in addition to the laborious *in vivo* pathotyping assay, alternative methods based on the molecular characteristics of *meq* oncogene sequences and the corresponding encoded Meq oncoprotein have been commonly used for phylogenetic analysis and virulence prediction of novel MDV-1 strains and have been published in numerous studies from various countries [12–14].

Despite the wide and routine application of vaccination, outbreaks of MD still occasionally occur in vaccinated chicken farms in numerous Asian countries, including China [15], India [16], Japan [17] and Thailand [18]. During the past 20 years, MD-related cases have frequently been found in chicken populations in Taiwan; however, the nearest published report of very virulent MDV-1 appearing and circulating among local chickens or layers in poultry flocks in Taiwan was before the 21st century [19]. The constant lack of continuous monitoring of the genotypes and virulence of the circulating MDV-1 strains in Taiwan has led MD prevention to become a thorny issue, which may result in inadequate responses to the sudden MD epidemic. In this study, we present the phylogenetic and virulence characteristics of current circulating MDV-1 strains in Taiwan through sequence analysis of the *meq* oncogene obtained from vaccinated chicken flocks from 2018 to 2020.

## Materials and methods

### Samples

From January 2018 to December 2020, the chicken cases pathologically diagnosed with MD suspect from the Animal Disease Diagnostic Center of the National Pingtung University of Science and Technology (NPUST) were included in this study. The submitted chickens, including layers and native chickens, had been vaccinated with commercial univalent or bivalent MDV vaccines. The gross lesion tissues from these MD suspect chickens were examined by PCR assay for MDV-1 detection [20,21] and then stored at -80°C for further gene analysis.

### Nucleic acid extraction and field virus detection

A total of 17 cases were randomly selected among the MDV-detected cases (Table 1). The nucleic acid was extracted from collected tissue samples by using TANBead® Nucleic Acid Extraction Kit (TANBEAD, Taiwan) following the manufacturer's instructions and stored at -20°C. All extracted nucleic acid samples were further examined for avian leukosis virus (ALV) [22] and reticuloendotheliosis virus (REV) [23]. In addition, each case was also detected

**Table 1. Information of 18 confirmed MDV-1 strains obtained from 17 chicken flocks in Taiwan.**

| Flock ID | Location | Year | Age (weeks) | Genetic line | Vaccine types | Pathogen | Strain |
|---|---|---|---|---|---|---|---|
| P107-008 | Changhua | 2018 | 20 | Hy-Line | CVI988 | MDV, CAV | TW/008/18 |
| P107-009 | Pingtung | 2018 | 17 | Hy-Line | CVI988 | MDV, CAV | TW/009/18 |
| P107-011 | Pingtung | 2018 | 19 | Hy-Line | CVI988 | MDV, CAV | TW/011/18 |
| P107-014 | NA | 2018 | 26 | Hy-Line | CVI988 | MDV | TW/014/18 |
| P107-023 | Pingtung | 2018 | 35 | Hy-Line | CVI988 | MDV, CAV | TW/023/18 |
| P108-109 | Tainan | 2019 | 31 | Native Chicken | CVI988 + HVT | MDV, CAV | TW/109/19 |
| P108-123 | Pingtung | 2019 | NA | Hisex | CVI988 + HVT | MDV | TW/123/19 |
| P108-133 | Pingtung | 2019 | 10 | Layer, NA | CVI988 + HVT | MDV | TW/133/19 |
| P108-141 | Chiayi | 2019 | 14, 18 | Hisex | CVI988 + HVT | MDV, IBV, CAV, IBDV | TW/141A/19 TW/141B/19 |
| P108-146 | Chiayi | 2019 | 19 | Layer, NA | CVI988 + HVT | MDV, CAV | TW/146/19 |
| P108-147 | Tainan | 2019 | 23 | Layer, NA | CVI988 + HVT | MDV | TW/147/19 |
| P108-148 | Kaohsiung | 2019 | 19 | Layer, NA | CVI988 + HVT | MDV, CAV | TW/148/19 |
| P108-149 | Chiayi | 2019 | 23 | Layer, NA | CVI988 + HVT | MDV, CAV | TW/149/19 |
| P109-003 | Taitung | 2020 | 27 | Hisex | CVI988 + HVT | MDV | TW/003/20 |
| P109-029 | Chiayi | 2020 | 8 | Hisex | CVI988 + HVT | MDV, NDV | TW/029/20 |
| P109-048 | Pingtung | 2020 | 18 | Hy-Line | CVI988 | MDV, FAV | TW/048/20 |
| P109-116 | Chiayi | 2020 | 29 | Hisex | CVI988 + HVT | MDV, IBV, CAV, MS, FPV | TW/116/20 |

NA: Not available. MDV: Marek's disease virus; CAV: Chicken anemia virus; IBV: Infectious bronchitis virus; IBDV: Infectious bursal disease virus; NDV: Newcastle disease virus; FAV: Fowl adenovirus; MS: *Mycoplasma synoviae*; FPV: Fowl poxvirus.

for the positivity of suspected common avian pathogens, such as Newcastle disease virus (NDV) [24], infectious bursa disease virus (IBDV) [25], infectious bronchitis virus (IBV) [26], chicken anemia virus (CAV) [25], Fowl adenovirus (FAV) [27], *Mycoplasma synoviae* (MS) [28], Fowl poxvirus (FPV) [29], etc.

## PCR for *meq* oncogene

The *meq* oncogene was amplified with primers EcoR-Q-for: GGTGATATAAAGACGATAGTCATG and EcoR-Q-rev: CTCATACTTCGGAACTCCTGGAG by conventional PCR to produce 1,625-bp DNA fragment as described previously [12].

## Cloning and sequencing

The amplified *meq* oncogene products were purified by the FavorPrep™ Gel purification Mini Kit (FAVORGEN® BIOTECH CORP., Taiwan) according to the manufacturer's instructions and were cloned into T-vector using the T&A Cloning Vector Kit (Yeastern Biotech Co., Ltd., Taiwan). After blue-white screening, the plasmid-transformed colony was picked and cultured to acquire *meq* gene-carried plasmids for sequencing. Consensus sequences of the *meq* oncogene, which were confirmed by Sanger sequencing, were further verified and assembled using BLAST alignment analysis. The obtained nucleotide sequences of *meq* oncogenes of Taiwanese MDV-1 strains were submitted to the GenBank database with the accession numbers OQ576796-OQ576813.

## Genetic analysis

A total of 37 selected *meq* oncogene sequences were retrieved from the GenBank database as references (Table) for comparison with the sequences of Taiwanese strains used in this study.

Nucleotide and amino acid identifications were conducted by alignment of Taiwanese strains and references using Clustal W software [30]. The phylogenetic tree was constructed by MEGA version X [31] software using neighbor-joining (NJ) algorithms under the Tamura-Nei model with 1,000 bootstrap replicates. The sequences of Meq oncoprotein of Taiwanese strains were compared with selected references to identify the specific substitution of deduced amino acids. Additionally, the proline content and the number of PPPP motifs within the Meq oncoprotein of Taiwanese strains were also evaluated.

### Ethics statement

The study was approved by the author's institution (Animal Disease Diagnostic Center of National Pingtung University of Science and Technology), and the animals used for necropsy also had the consent of their owners. In addition, this study did not involve live animal experiments and non-human primate test subjects, so there are no relevant details about experimental animal.

## Results

### Profiles of collected samples

From 2018 to 2022, MDV-1 detection rates of the submitted chicken cases were 7.6%, 4.6%, 3.24%, 2.2%, and 2.2%, respectively. The information and the status of coinfection with other avian diseases of the randomly selected 17 cases are shown in Table 1. Interestingly, two MDV-1 strains, i.e., TW/141A/19 and TW/141B/19 in our collected materials, were detected from the same chicken flocks, indicating that different MDV-1s could simultaneously exist in identical populations. The presence of other poultry pathogens in the examined samples, along with MDV, indicates that pathogen coinfections in chicken flocks occur frequently nowadays in Taiwan. Notably, no positive detection of oncogenic virus ALV and REV were observed within all MD-positive materials. The chicken cases in this study mainly showed lymphoma lesions in a variety of organs and tissues, such as the ovary, lung, heart, mesentery, kidney, liver, spleen, thymus, pancreas, proventriculus, intestine, and skeletal muscle, and a few of them had neuronal lesions, indicating that the visceral lymphoma of MD was of a significant epidemic form (Fig 1) rather than ALV or REV.

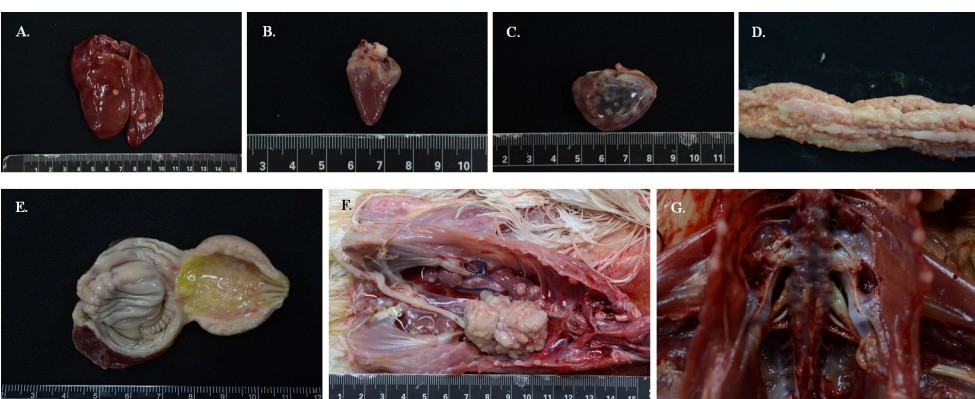

**Fig 1. The gross lesions of MDV-1 clinical cases.** The gross lesions in MDV-1 infected chickens in this study include: Enlarged liver (A) with white neoplastic nodules; the variable size of multifocal grayish-white nodules in the heart (B) and spleen (C); numerous white nodules throughout the intestinal serosa surface (D); thickened proventricular wall with multiple white protrusion (E); multifocal white nodules in the kidney (F); neoplastic mass occupied ovary (F); enlarged left sacrum nerve plexus (G).

## Phylogenetic analysis of meq oncogenes of Taiwanese MDV-1 strains

A total of 18 MDV-1 strains were obtained from 17 vaccinated chicken flocks in Taiwan between 2018 and 2020. The genetic features of these 18 obtained *meq* oncogenes were characterized through phylogenetic analysis with 37 selected reference genes of identified strains, which were collected from various locations and available in the GenBank database. Among the selected reference strains, 25 of the strains were pathotyped. The phylogenetic tree demonstrated that the analyzed *meq* oncogenes in this study could be separated into 4 clusters (Fig 2). Cluster 1 involved all Chinese strains, Thai strains, Taiwanese strains, and some of the Japanese strains. The vaccine strains, mild virulent strains, and Australian strains were all included in Cluster 2. Most classic USA strains representing pathotypes of very virulent and very virulent plus were grouped into Cluster 3, whereas part of the USA strains were divided into Cluster 4 with strains from India and Japan. The homology range among the members of Cluster 1 was 99.3–100% nucleotide identity and 98.5–100% amino acid identity, respectively. Notably, five of 18 Taiwanese strains showed the closest relationship with the vvMDV strain LS of China and 4 recently identified strains of Thailand (100% nucleotide and amino acid identity, respectively). In addition, the frequent appearance of branches from 13 Taiwanese strains in Cluster 1 indicated a high probability of individual evolution of MDV-1. These results suggested the circulation of MDV-1 for a particular duration among chicken flocks in Taiwan, which brought about geographical genetic polymorphism.

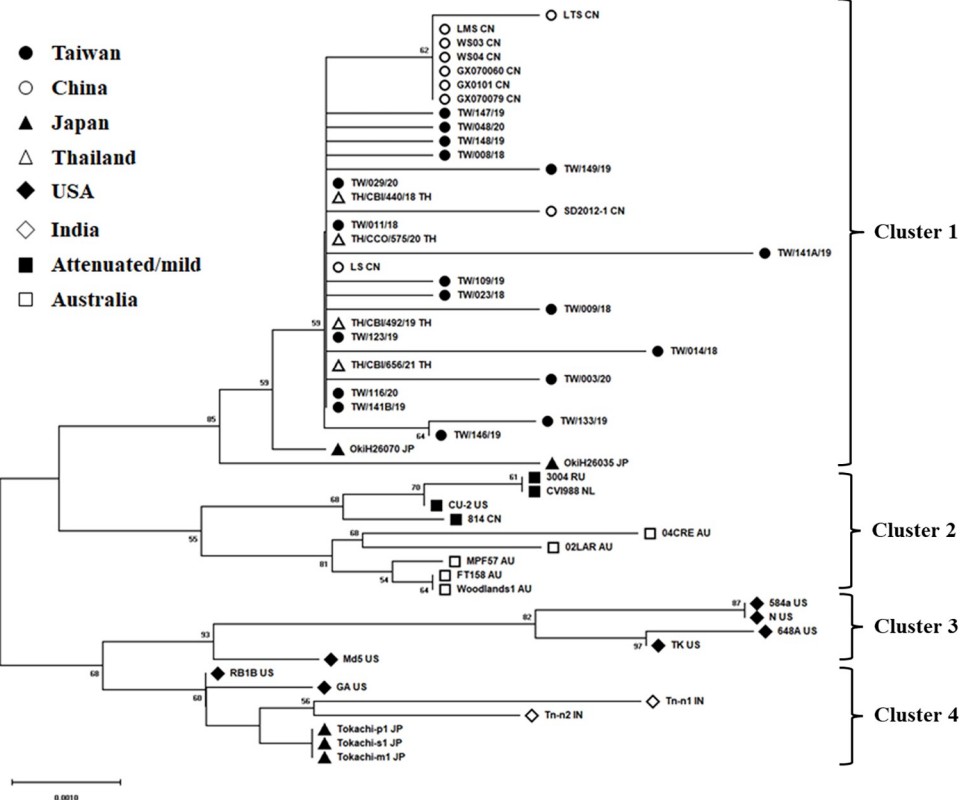

**Fig 2. Phylogenetic tree of MDV-1 strains.** The phylogenetic tree was built by using neighbor-joining (NJ) based on the complete nucleotide sequences of *meq* oncogene obtained from reference MDV-1 strains and Taiwan field strains. All reference strain names are labeled with the corresponding abbreviation of countries. The symbols indicate respective field MDV-1 strains in different countries and the attenuated/mild strains.

## Molecular characterization of Meq oncoproteins of Taiwanese MDV-1 strains

The deduced amino acid sequences were compared with known pathotyping strains from different countries to investigate the characterization of Meq oncoproteins of Taiwanese isolations (Table 2). Specific amino acid substitutions, including positions 71, 77, 80, 88, 93, 110, 115, and 119 in the basic-leucine-zipper domain and 139, 153, 176, 180, 217, 218, 277, 283, and 326 in the transactivation domain of the Meq oncoprotein, were verified previously to be correlated with MDV-1 virulence [10,18]. All of the Meq oncoproteins of Taiwanese strains shared identical substitutions at positions 71(A), 77(E), 80(Y), 115(A), and 176(R) with the vvMDV strains LS and GX0101 [32,33], and the vv+MDV strains LTS and SD2012-1 [34,35]

**Table 2. Specific amino acid substitutes in Meq oncoprotein of presented Taiwan MDV-1 field strains compared with the reference strains.**

| Strain | Pathotype | 71 | 77 | 80 | 88 | 93 | 110 | 115 | 119 | 139 | 153 (PPPP) | 176 (PPPP) | 180 | 217/276[b] (PPPP) | 218/277[b] (PPPP) | 277/336[b] | 283/342[b] | 326/385[b] |
|---|---|---|---|---|---|---|---|---|---|---|---|---|---|---|---|---|---|---|
| 648A (USA) | vv+MDV | A | K | D | A | Q | C | V | R | T | Q | A | A | A | P | P | A | T |
| N (USA) | vv+MDV | A | K | D | A | Q | C | V | R | T | Q | A | A | A | P | P | A | T |
| Md5 (USA) | vvMDV | A | K | D | A | Q | C | V | C | T | P | P | T | A | P | L | V | T |
| RB1B (USA) | vvMDV | A | K | D | A | Q | C | V | C | T | P | P | T | P | P | L | A | T |
| GA (USA) | vMDV | A | K | D | A | Q | C | V | C | T | P | P | T | P | P | L | A | T |
| 571 (USA) | vMDV | A | E | D | A | Q | C | V | C | T | P | H | T | P | P | L | A | T |
| CU-2 (USA) | mMDV | S | E | D | A | Q | S | V | C | T | P | P | T | P | P | L | A | I |
| CVI988/Rispens (NL) | attMDV | S | E | D | A | Q | S | V | C | T | P | P | T | P | P | L | A | I |
| SD2012-1 (CN) | vv+MDV | A | E | Y | T | Q | C | A | C | T | P | R | T | A | P | L | A | T |
| LTS | vv+MDV | A | E | Y | A | Q | C | A | C | A | P | R | T | A | P | L | A | T |
| GX0101 (CN) | vvMDV | A | E | Y | A | Q | C | A | C | A | P | R | T | A | P | L | A | T |
| LS (CN) | vvMDV | A | E | Y | A | Q | C | A | C | T | P | R | T | A | P | L | A | T |
| TH/CBI/656/21 (THA) | NA[a] | A | E | Y | A | Q | C | A | C | T | P | R | T | A | P | L | A | T |
| OkiH26035 (JP) | NA | A | E | Y | A | Q | C | A | C | T | P | P | T | A | P | L | A | T |
| TW/008/18 | NA | A | E | Y | A | Q | C | A | C | T | P | R | T | A | P | L | A | T |
| TW/009/18 | NA | A | E | Y | A | Q | C | A | C | T | P | R | T | A | P | L | A | T |
| TW/011/18 | NA | A | E | Y | A | Q | C | A | C | T | P | R | T | A | P | L | A | T |
| TW/014/18 | NA | A | E | Y | A | Q | C | A | C | T | P | R | T | A | P | L | A | T |
| TW/023/18 | NA | A | E | Y | A | Q | C | A | C | T | P | R | T | A | P | L | A | T |
| TW/109/19 | NA | A | E | Y | A | Q | C | A | C | T | P | R | T | A | P | L | A | T |
| TW/123/19 | NA | A | E | Y | A | Q | C | A | C | T | P | R | T | A | P | L | A | T |
| TW/133/19 | NA | A | E | Y | A | Q | C | A | C | T | P | R | T | A | P | L | A | T |
| TW/141A/19 | NA | A | E | Y | A | Q | C | A | C | T | P | R | T | A | P | L | A | T |
| TW/141B/19 | NA | A | E | Y | A | Q | C | A | C | T | P | R | T | A | P | L | A | T |
| TW/146/19 | NA | A | E | Y | A | Q | C | A | C | T | P | R | T | A | P | L | A | T |
| TW/147/19 | NA | A | E | Y | A | Q | C | A | C | T | P | R | T | A | P | L | A | T |
| TW/148/19 | NA | A | E | Y | A | Q | C | A | C | T | P | R | T | A | P | L | A | T |
| TW/149/19 | NA | A | E | Y | A | Q | C | A | C | T | P | R | T | A | P | L | A | T |
| TW/003/20 | NA | A | E | Y | A | Q | C | A | C | T | P | R | T | A | P | L | A | T |
| TW/029/20 | NA | A | E | Y | A | Q | C | A | C | T | P | R | T | A | P | L | A | T |
| TW/048/20 | NA | A | E | Y | A | Q | C | A | C | T | P | R | T | A | P | L | A | T |
| TW/116/20 | NA | A | E | Y | A | Q | C | A | C | T | P | R | T | A | P | L | A | T |

[a] NA: Not available.

[b] Amino acid position based on 59-a.a. insertion-containing Meq oncoprotein which is predominantly in lower virulence strain.

**Table 3. The proline content and the number of 4-proline-repeat (PPPP) within Meq oncoproteins in the presented Taiwan MDV-1 field strains and reference strains.**

| Strain | Pathotype | Size of Meq (a. a.) | Insertion size (a. a.) | Proline contents (%) | Number of PPPPs |
|---|---|---|---|---|---|
| CVI988/Rispens (NL) | attMDV | 398 | 59 | 23.1 | 7 |
| CU-2 (USA) | mMDV | 398 | 59 | 23.1 | 7 |
| 648A (USA) | vv+MDV | 339 | Nil[b] | 20.9 | 2 |
| N (USA) | vv+MDV | 339 | Nil | 20.9 | 2 |
| Md5 (USA) | vvMDV | 339 | Nil | 21.3 | 4 |
| RB1B (USA) | vvMDV | 339 | Nil | 21.5 | 5 |
| GA (USA) | vMDV | 339 | Nil | 21.5 | 5 |
| 571 (USA) | vMDV | 339 | Nil | 21.2 | 4 |
| SD2012-1 (CN) | vv+MDV | 339 | Nil | 20.9 | 3 |
| LTS (CN) | vv+MDV | 339 | Nil | 20.9 | 3 |
| GX0101 (CN) | vvMDV | 339 | Nil | 20.9 | 3 |
| LS (CN) | vvMDV | 339 | Nil | 20.9 | 3 |
| TH/CBI/656/21 (THA) | NA[a] | 339 | Nil | 20.9 | 3 |
| OkiH26035 (JP) | NA | 339 | Nil | 21.2 | 4 |
| TW/008/18 | NA | 339 | Nil | 20.9 | 3 |
| TW/009/18 | NA | 339 | Nil | 20.9 | 3 |
| TW/011/18 | NA | 339 | Nil | 20.9 | 3 |
| TW/014/18 | NA | 339 | Nil | 20.6 | 3 |
| TW/023/18 | NA | 339 | Nil | 20.9 | 3 |
| TW/109/19 | NA | 339 | Nil | 20.9 | 3 |
| TW/123/19 | NA | 339 | Nil | 20.9 | 3 |
| TW/133/19 | NA | 339 | Nil | 20.6 | 3 |
| TW/141A/19 | NA | 339 | Nil | 20.9 | 3 |
| TW/141B/19 | NA | 339 | Nil | 20.9 | 3 |
| TW/146/19 | NA | 339 | Nil | 20.9 | 3 |
| TW/147/19 | NA | 339 | Nil | 20.9 | 3 |
| TW/148/19 | NA | 339 | Nil | 20.9 | 3 |
| TW/149/19 | NA | 339 | Nil | 20.9 | 3 |
| TW/003/20 | NA | 339 | Nil | 21.3 | 3 |
| TW/029/20 | NA | 339 | Nil | 20.9 | 3 |
| TW/048/20 | NA | 339 | Nil | 20.9 | 3 |
| TW/116/20 | NA | 339 | Nil | 20.9 | 3 |

of China. Although genetic analysis demonstrated the existence of high virulence MDV-1 strains in Taiwan, four unique substitution positions 119(R), 153(Q), 176(A) and 277(P) were not present when comparing with the classic vv+MDV strains N and 648A of USA. Moreover, as in previous reports, the proline content and the number of PPPP repeats within the Meq oncoprotein were also used as virulence predictors for Taiwanese strains [11,36]. Compared with vaccine and mild MDV-1 strains (Table 3), the Taiwanese strains lacked insertions and showed related lower proline contents as well as PPPP motif numbers, which supported the high virulence prediction.

## Discussion

This is the first report of MDV-1 virulence by molecular analyses in nearly 20 years after the study on the polymorphism of MDV-1 strains and the presence of vvMDV in Taiwan [19].

The present study revealed the occurrence and genetic properties of the MDV-1 field strains circulating in Taiwan based on the sequence analysis of 18 virulence-associated *meq* oncogenes obtained from 17 vaccinated chicken flocks collected during 2018–2020. Therefore, understanding the genetic characterization of Taiwan MDV-1 has become a primary concern for disease prevention and control.

Phylogenetic analysis demonstrated that all Taiwanese strains were grouped into the same cluster, involving predominantly highly virulent MDV-1 strains from China and field strains from Thailand and Japan. Some Taiwanese strains showed complete genetic identity to the LS strain, which was isolated from the Sichuan province of China and classified as the vvMDV pathotype [33]. High similarity features are also represented in Thai field strains, which were recently published as being in close phylogenetic relationship with MDV-1 strains from China [18], indicating that these field MDV-1 strains may share a common ancestor and evolutionary direction. Interestingly, Guangxi Province is geographically closer to Taiwan and Thailand than to Sichuan Province; however, based on phylogenetic analysis, the strains from Guangxi, GX070060 and GX070079, showed less phylogenetic relationships with Taiwanese and Thai strains. The reasons of these findings are still unknown, but the possibility of pathogen transmission by wild birds could be considered [37]. In the present study, the 'LS-like' MDV-1 field strains, including TW/011/18, TW/123/19, TW/141B/19, TW/029/20, and TW/116/20, were obtained from different flocks in various collecting years, indicating that these strains were dominant stains persistently circulating in chicken farms in Taiwan. The persistent detection of such strains from vaccinated flocks might be due to the genetic adaptation in the chicken flocks and farms and the immune escape from the vaccine protection [38].

Taiwanese MDV-1 strains were all clustered together in Cluster 1 of the phylogenetic tree and spread in several different branches, which revealed not only geographically restricted evolution, but also the genetic diversity as in previous investigations [39,40]. Notably, the strains from Southern Japan were grouped into the cluster with Taiwanese MDV-1 and Chinese strains, whereas the Northern Japanese strains were clustered into another group with USA and Indian strains, suggesting a possible independent construction of geographical phylogeny in East Asia.

The spontaneous mutations of oncogenes, especially the *meq* oncogene, on the MDV-1 genome have been regarded as important roles corresponding to increasing virulence [41]. The Meq oncoprotein, known to play a critical role in MDV-1 pathogenicity, has shown unexpectedly higher mutation rates than general DNA viruses and even resembles RNA viruses [42]. Although the causes for such high mutation frequency of MDV-1 have not been fully clarified, most investigations have demonstrated that the improper use of vaccines can lead to the induction of positive selection from the field viruses, eventually resulting in viral diversity [43,44]. With the annually found MD clinical cases and the genetic diversity of *meq* oncogenes in our results (Fig 2, Cluster 1), the positive selection of the viruses in vaccinated chicken flocks of Taiwan may drive the viral evolved direction toward enhanced virulence of MDV-1.

Specific sequence characterization of the Meq oncoprotein has been reported as a predictor for MDV-1 pathotype and can be applied to the virulence prediction for novel isolated MDV strains instead of *in vivo* classification [16,18,36]. It has been reported previously that amino acid mutations at positions 71 (Ala), 77 (Glu), 80 (Tyr), 115 (Ala), and 176 (Arg) were the main feature of highly virulent MDV-1 of Chinese strains [14,17]. The results of sequence analyses in our study showed that all obtained Taiwanese MDV-1 field strains represented the molecular characteristics of the mutations as the previous report of China strains, supporting the high virulent potential of these Taiwanese MDV-1 strains. In addition, the mutations at positions 77, 80, 115, and 176 of Meq oncoproteins seem to be common features of Chinese, Thai, Japanese, and Taiwanese MDV-1 field strains, and could be considered as accessible markers for molecular identification of East and Southeast Asian MDV-1 strains.

Insertions appearing in *meq* oncogenes of mild and attenuated strains, such as CU-2 and CVI988/Rispens, cause the expression of longer Meq oncoproteins, resulting in the presence of higher proline contents and more PPPP motifs than those of virulent MDV-1 strains which were correlated with low virulence characteristics of MDV strains. Conversely, no insertions in *meq* oncogenes of N and 648A strains of USA have lower proline contents and fewer PPPP motifs, leading to high virulence MDV strains [11]. Our findings in the present study showed the related lower proline contents and fewer PPPP motifs, and the values were between those of vvMDV and vv+MDV USA strains. In addition, the related lower proline contents and more occasional PPPP motifs of Taiwanese strains were similar to the values of vvMDV and vv +MDV Chinese strains. These results indicated that the circulating MDV-1 field strains in Taiwan were potentially hypervirulent, but their exact pathotypes still required further classification by *in vivo* pathotyping experiments.

It is still a vital and effective way to control MDV epidemics using vaccines in flocks [4]. In Taiwan, vaccination programs for young chickens via bivalent vaccines of two commercial live strains CVI988/Rispens of MDV-1 and FC126 of HVT, have been commonly practiced across the poultry industries. To the best of our knowledge, bivalent vaccination is available for producing a protective immune response against most virulent MDVs, including vvMDV and vv +MDV pathotypes, but the occurrence of clinical MD cases due to immune failure in chicken flocks around the world, including in Taiwan, which raising close attention to the problems regarding the vaccine application. Current commercial MD vaccines are all cell-associated types with more transportation, storage, and administration difficulties than other live vaccines. Vaccination efficiency can be affected by the reconstituted conditions, performance, dose uniformity of vaccines, etc. [45,46]. In Taiwan, we have examined the immune status by detecting MDV from feather tips 14–21 post-vaccination day after the pullets were applied to the CVI988 and/or HVT-FC126 on 1 day of age. Only 48% (16/33) of chicken flocks were vaccinated successfully (achieving 70% immunization coverage). After monitoring the 7 flocks from a layer breeding farm, in which the pullets were vaccinated by applying the same patch of CVI988 vaccine, and the same injection machine and procedure were used, various detection rates of the vaccinated virus in the 7 flocks were found (30–90%) [47]. Ununiformed vaccine doses received by pullets were considered the possible reason for the uneven vaccination efficacy, and applying the well-mixed vaccines was essential to prevent immune failure.

Coinfection of avian viruses, such as MDV, IBDV, NDV, CAV, reovirus, and reticuloendotheliosis virus, can induce immunosuppression in infected hosts, reducing vaccination efficiency [48,49]. The coexistence of poultry immunosuppressive disease virus together with MDV has been detected in the present study, suggesting that the chicken flocks in Taiwan may also suffer under immune suppression and cannot have proper protection after vaccination.

In conclusion, the phylogenetic findings on the geographical diversity of *meq* oncogenes suggested an ongoing evolution in circulating Taiwanese MDV-1 strains, which already adapted to the chicken farms in Taiwan. The circulation of field MDV-1 strains in Taiwan was dominated by a cluster with potentially hypervirulent characterization. Routine surveillance of field MDV-1 strains and monitoring of immune status on poultry farms will be needed to develop effective vaccines and control strategies in response to the emergence of enhanced virulent Taiwanese MDV-1 strains.

## Supporting information

**S1 Table. Profile list of MDV-1 strains used in this study.**
(DOCX)

**S1 File.**
(DOCX)

## Acknowledgments

We appreciate the support of the Animal Disease Diagnostic Center of National Pingtung University of Science and Technology for providing clinical samples and co-operation in pathogenic diagnosis and virus detection.

## Author Contributions

**Conceptualization:** Ming-Chu Cheng, Yi-Yang Lien.

**Formal analysis:** Ming-Chu Cheng, Guan-Hua Lai, Yi-Lun Tsai.

**Funding acquisition:** Ming-Chu Cheng.

**Methodology:** Ming-Chu Cheng, Guan-Hua Lai.

**Resources:** Ming-Chu Cheng, Yi-Yang Lien.

**Writing – original draft:** Ming-Chu Cheng, Guan-Hua Lai.

**Writing – review & editing:** Ming-Chu Cheng, Guan-Hua Lai, Yi-Lun Tsai, Yi-Yang Lien.

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
