## [Decision Letter · Decision Letter 0]

5 Mar 2024

PONE-D-23-40205Circulating hypervirulent Marek’s disease viruses in vaccinated chicken flocks in Taiwan by genetic analysis of meq oncogenePLOS ONE

Dear Dr. Cheng,

Thank you for submitting your manuscript to PLOS ONE. After careful consideration, we feel that it has merit but does not fully meet PLOS ONE’s publication criteria as it currently stands. Therefore, we invite you to submit a revised version of the manuscript that addresses the points raised during the review process.

We look forward to receiving your revised manuscript.

Kind regards,

Vishwanatha R. A. P. Reddy

Academic Editor

PLOS ONE

2. PLOS requires an ORCID iD for the corresponding author in Editorial Manager on papers submitted after December 6th, 2016. Please ensure that you have an ORCID iD and that it is validated in Editorial Manager. To do this, go to ‘Update my Information’ (in the upper left-hand corner of the main menu), and click on the Fetch/Validate link next to the ORCID field. This will take you to the ORCID site and allow you to create a new iD or authenticate a pre-existing iD in Editorial Manager. Please see the following video for instructions on linking an ORCID iD to your Editorial Manager account: " ext-link-type="uri" xlink:type="simple">https://www.youtube.com/watch?v=_xcclfuvtxQ".

3. Thank you for uploading your study's underlying data set. Unfortunately, the repository you have noted in your Data Availability statement does not qualify as an acceptable data repository according to PLOS's standards.

4. We note that Figure 1 in your submission contain copyrighted images. All PLOS content is published under the Creative Commons Attribution License (CC BY 4.0), which means that the manuscript, images, and Supporting Information files will be freely available online, and any third party is permitted to access, download, copy, distribute, and use these materials in any way, even commercially, with proper attribution. For more information, see our copyright guidelines: http://journals.plos.org/plosone/s/licenses-and-copyright.

5. Please include your tables as part of your main manuscript and remove the individual files. Please note that supplementary tables (should remain/ be uploaded) as separate ""supporting information"" files.

Reviewers' comments:

Reviewer's Responses to Questions

**Comments to the Author**

1. Is the manuscript technically sound, and do the data support the conclusions?

Reviewer #1: Yes

Reviewer #2: Yes

2. Has the statistical analysis been performed appropriately and rigorously? 

Reviewer #1: N/A

Reviewer #2: Yes

3. Have the authors made all data underlying the findings in their manuscript fully available?

Reviewer #1: Yes

Reviewer #2: Yes

4. Is the manuscript presented in an intelligible fashion and written in standard English?

Reviewer #1: Yes

Reviewer #2: Yes

5. Review Comments to the Author

Reviewer #1: � Did the authors check the co-infection with ALV and REV in these cases? If yes, they can give the status of these oncogenic co-infections with MD, as in many cases, co-infection of these two predisposes the MD outbreak.

Authors should have studied other oncogenes such as pp38, vIL-8 and PP38 also, why these oncogenes are not investigated as this sequence information would be useful

No isolation was done in this work, so authors should mention as MDV strains rather than isolates in their manuscript. Also, how 18 MDV were obtained from 17 cases?

Reviewer #2: English language Writing of the manuscript must be improved before publication.

Results section could be improved and more detailed supporting evidences can be provided to support the conclusions.

Would be great if the authors could evaluate the isolates in vivo to support their hypothesis.

6. PLOS authors have the option to publish the peer review history of their article (what does this mean?). If published, this will include your full peer review and any attached files.

Reviewer #1: No

Reviewer #2: **Yes: **Muhammad Abid

---

## [Author Response · Author response to Decision Letter 0]

19 Apr 2024

We thank for the opportunity to submit our revised draft of the manuscript titled “Circulating hypervirulent Marek’s disease viruses in vaccinated chicken flocks in Taiwan by genetic analysis of meq oncogene” for the publication in PLoS One journal. We appreciated the time and efforts that you and the reviewers devote to providing feedback about the manuscript along with insightful comments and valuable improvements to this study.

---

## [Editor Report · Decision Letter 1]

24 Apr 2024

Circulating hypervirulent Marek’s disease viruses in vaccinated chicken flocks in Taiwan by genetic analysis of meq oncogene

PONE-D-23-40205R1

Dear Dr. Cheng,

We’re pleased to inform you that your manuscript has been judged scientifically suitable for publication and will be formally accepted for publication once it meets all outstanding technical requirements.

Kind regards,

Vishwanatha R. A. P. Reddy

Academic Editor

PLOS ONE

---

## [Editor Report · Acceptance letter]

29 Apr 2024

PONE-D-23-40205R1 

PLOS ONE

Dear Dr. Cheng, 

I'm pleased to inform you that your manuscript has been deemed suitable for publication in PLOS ONE. Congratulations! Your manuscript is now being handed over to our production team.

Kind regards, 

on behalf of

Dr. Vishwanatha R. A. P. Reddy 

Academic Editor

PLOS ONE